# Association between Poverty and Refraining from Seeking Medical Care during the COVID-19 Pandemic in Japan: A Prospective Cohort Study

**DOI:** 10.3390/ijerph20032682

**Published:** 2023-02-02

**Authors:** Erika Obikane, Daisuke Nishi, Akihiko Ozaki, Tomohiro Shinozaki, Norito Kawakami, Takahiro Tabuchi

**Affiliations:** 1Department of Social Medicine, National Center for Child Health and Development, Tokyo 157-0074, Japan; 2Department of Mental Health, Graduate School of Medicine, The University of Tokyo, Tokyo 113-0033, Japan; 3Department of Public Mental Health Research, National Institute of Mental Health, National Center of Neurology and Psychiatry, Tokyo 187-8553, Japan; 4Department of Breast and Thyroid Surgery, Jyoban Hospital of Tokiwa Foundation, Iwaki 972-8322, Japan; 5Department of Information and Computer Technology, Faculty of Engineering, Tokyo University of Science, Tokyo 162-8601, Japan; 6Department of Digital Mental Health, Graduate School of Medicine, The University of Tokyo, Tokyo 113-8655, Japan; 7Department of Cancer Control Center, Osaka International Cancer Institute, Osaka 541-8567, Japan; 8The Tokyo Foundation for Policy Research, Tokyo 106-6234, Japan

**Keywords:** low household income, relative poverty, inequality, access to medical care, COVID-19

## Abstract

This limited study examined how low household income affected avoidant behaviors to seek medical care during the pandemic. We investigated an association between household income below the relative poverty line and refraining from seeking medical care (RSMC) in a longitudinal study during the COVID-19 pandemic. We conducted an analysis of a population-based internet cohort in Japan. Individuals aged 20 to 79 years old living in Japan participated in the internet surveys between 2020 and 2021. The primary outcome was the RSMC of regular visits and new symptoms in 2021. A total of 19,672 individuals were included in the analysis. Household income below the relative poverty line in 2020 was significantly associated with refraining from seeking regular medical visits for men and women (for men, odds ratio: 1.28; confidence interval: 1.19, 1.83; for women, odds ratio: 1.42; confidence interval: 1.14, 1.82) in 2021, after accounting for RSMC in 2020. Relative poverty in 2020 was also associated with the RSMC of new symptoms among men (for males, odds ratio: 1.32; confidence interval: 1.05, 1.66) in 2021 after adjusting for covariates. The study suggested the need to alleviate the financial burden of vulnerable people seeking medical care and advocate for making necessary medical visits, even in a pandemic.

## 1. Introduction

Delayed access to medical care is often preceded by refraining from seeking medical care (RSMC) [1]. Previous studies reported that people avoided medical care due to traditional barriers to medical care, such as high medical costs, lack of insurance and time constraints [2,3,4,5], and decreased perceived needs and unfavorable evaluations for seeking medical care [6,7].

The COVID-19 pandemic impacted the lives of people in the world. Japan was not an exception, experiencing several COVID-19 outbreaks since January 2020. The Japanese government declared a state of emergency in all prefectures in Japan in April 2020, followed by a second state of emergency in 11 prefectures in January 2021; however, the extent to which the pandemic’s influence on people’s help-seeking behaviors in medicine differed according to income and fear is still unknown.

Previous studies before the COVID-19 outbreaks have shown that there were income inequalities in medical care utilization [8,9]. Moreover, these income disparities in medical care utilization are known to be moderated by certain characteristics, such as sex [10,11,12,13], living areas or countries [14,15], or chronic conditions [16]. A previous U.S. study showed that people with low household income were more likely to present delayed medical care use than their counterparts with higher income during the COVID-19 pandemic [17]; however, how income inequalities in medical care utilization during the pandemic are exaggerated by sex, chronic conditions, or living areas should be explored in depth. Not only will these findings clarify the impact of low household income on RSMC, but they will also give us a clue as to who are the more vulnerable ones among those with low household income, as well as what type of medical visits will be influenced the most. These findings may lead us to more specific and practical implications (i.e., providing more monetary or nonmonetary support for making medical visits for people with underlying diseases and low income, etc.) for overcoming income disparities in health care utilization during the COVID-19 pandemic.

Fear of COVID-19 is another factor that may lead to people’s avoidance in seeking medical care in the pandemic. Previous studies reported that the fear of infection caused delayed medical care utilization [18,19] or decreased medical care utilization [20,21]. Studies are needed to investigate the effect of fear of COVID-19 on RSMC, as well as to consider the effect of fear of COVID-19 in evaluating the association between income and RSMC during the pandemic.

Our research questions are as follows: How does low household income influence the RSMC of regular and new visits during the pandemic, even after adjusting for fears of COVID-19? How is the impact of low household income on RSMC different depending on demographic or social factors? The study aim is to investigate the association between household income below the relative poverty line in 2020 and the RSMC of regular visits and new symptoms in 2021, after accounting for RSMC in 2020, using a large internet longitudinal study. The study also evaluates the association between the fear of COVID-19 and the RSMC of regular and new visits in 2020 and 2021. The study then discusses differences in sex, underlying diseases, and living areas with or without a specific alert on the association between low household income on the RSMC of regular and new visits. The findings of the study may contribute to policy implications for promoting appropriate medical visits among vulnerable population during the COVID-19 pandemic.

## 2. Materials and Methods

### 2.1. Data Sources

We analyzed longitudinal datasets using the Japan COVID-19 and Society Internet Survey (JACSIS) in 2020 and 2021. The JACSIS was a prospective cohort survey aimed to investigate the impact of the COVID-19 pandemic on social and health conditions, and the details are described in the previous study [22]. The two surveys used in the study were conducted between August and September 2020 and February 2021, respectively. These surveys were managed by a nationwide internet research company, Rakuten Insight, with 2.3 million registered candidates with demographic information. Candidates were randomly selected and were asked to participate in the survey.

### 2.2. Participants

The study included participants aged 20–69 years old, who completed surveys in both 2020 and 2021. The original dataset included participants aged 15–69 years old, but as participants aged 15–19 years old were considered minors in Japan and were more likely to be economically dependent on their guardians, they were excluded from the study. To maintain the quality of the data, this study excluded inconsistent responses in the screening questions. The following criteria were used to assess the inconsistency: (1) an invalid response for not choosing the right alternative as specified (respondents were asked to choose the second option from the bottom); (2) participants checking all the items to the questions regarding drug use (i.e., marijuana, cocaine, or heroin); and (3) checking all the items for having 16 underlying chronic diseases. The algorithms of excluding inconsistent responses are described in detail in the previous study [22].

### 2.3. Variable Definitions

Exposure. The study used equivalent household annual income per capita in 2020 as the primary exposure variable. The equivalent household annual income per capita was calculated by annual household income divided by the square root of household size. The equivalent annual income per capita was categorized as household income below relative poverty line (household annual income below 50% of median income; <3 million yen); household income above relative poverty line (household annual income at or above 50% of median income; >=3 million yen); and no response. The study used the fear of COVID-19 as the secondary exposure variable, measured by measured by the Japanese version of the Fear of Coronavirus-19 Scale, with a higher score indicating a higher level of fear of the COVID-19 infection. The sum of all the items was categorized into 7–15, 16–20, 21–25, and 26–35 points according to the previous studies [23,24].

Outcomes. As outcome variables, we measured the RSMC of regular medical visits or RSMC of new symptoms at two time points during the COVID-19 pandemic, between August to September in 2020 and February in 2021. The RSMC of regular medical visits was identified by the question: “Did you refrain from seeking regular medical visits in the past 2 months?” The RSMC of new symptoms was identified by the following question: “Did you refrain from seeking medical visits for new symptoms in the past 2 months?” The responses were recorded as yes or no.

Covariates. The study used the covariates, including variables that were used in the previous studies on RSMC [8,9,17,18,20,21], as well as the additional variable (fears of COVID-19) that was considered important during the pandemic. For covariates, the study used age (20–34; 35–49; 50–64; and 65–79 years), educational attainment (junior high school or high school; technical or junior college or college dropout; college or university or graduate school; others or missing), employment status (full or part time; unemployed; others or missing), and fear of COVID-19 from the dataset in 2020. The study also conducted a stratified analysis according to whether living areas of the participants were subject to the state of emergency or specific alert for the COVID-19 at the time of data collection in 2020 or 2021. Alert specific areas in 2020 referred to restricted areas subject to the state of emergency or specific alert prefectures in April 2020 (Tokyo, Kanagawa, Chiba, Osaka, Hyogo, Fukuoka, Hokkaido, Ibaragi, Ishikawa, Gifu, Aichi, and Kyoto). Alert specific areas in 2021: restricted areas subject to the state of emergency in February 2021 (Tokyo, Kanagawa, Chiba, Saitama, Tochigi, Aichi, Gifu, Osaka, Kyoto, Hyogo, and Fukuoka).

### 2.4. Statistical Analyses

We first calculated the number and proportion of participants presenting each demographic characteristics by equivalent annual income and sex. We conducted a logistic regression to calculate an odds ratio of household income below the relative poverty line for the RSMC of regular visits in the year 2020 and 2021 by sex, after accounting for age, educational attainment, employment status, fear of COVID-19, and RSMC in 2020 (for RSMC in 2021 only). We also calculated an odds ratio of household income below the relative poverty line for the RSMC of new symptoms in the year 2020 and 2021 by sex, after adjusting for the covariates. For all models, we assessed whether the adjusted ORs between relative poverty and RSMC in 2021 vary across the fear of COVID-19 categories by including their interaction terms; as we did not find any difference in ORs between fear of COVID-19, we only included the main terms of the variables. We then conducted a logistic regression to compute an odds ratio of the fear of COVID-19 for the RSMC of regular visits in the year 2021 by sex, after adjusting for age, educational attainment, employment status, equivalent annual income in 2020, and RSMC in 2020. We conducted a similar analysis to compute an odds ratio of the fear of COVID-19 for the RSMC of new symptoms in 2021, adjusting for the covariates. In the subgroup analysis, we calculated an adjusted odds ratio of household income below the relative poverty line for RSMC in 2020 and 2021 among participants who reported to have any underlying medical conditions, including hypertension, diabetes, respiratory diseases, cardiovascular diseases, stroke, cancer, or mental disorders. We also calculated odds ratios of household income below the relative poverty line for RSMC in 2020 and 2021 according to alert specific areas for the COVID-19 pandemic. A *p*-value of 0.05 was considered significant. Statistical analysis was conducted using the SAS version 9.4 (SAS Institute, Cary, NC, USA).

## 3. Results

Of the 22,840 participants who responded to both the year 2020 and 2021 surveys, 2501 participants who presented inconsistent responses in screening questions were excluded from the study. We also excluded 667 participants aged 15–19 years from the database, and 19,672 participants remained in the analysis. The characteristics of the study participants, numbers, and proportions of participants presenting the RSMC of regular or new medical visits according to equivalent annual income and sex are summarized in Table 1. The proportions of participants with an RSMC of regular medical visits in 2021 accounted for 4.5% among males and 5.5% among females, respectively. The proportions of participants’ RSMC of new symptoms in 2021 accounted for 6.1% among males and 9.0% among females, respectively.

The odds ratios (ORs) of equivalent annual income for the RSMC of regular medical visits are summarized in Table 2. Both male and female participants with equivalent annual income below the relative poverty line presented a significantly higher OR for the RSMC of regular medical visits in 2021 (adjusted OR 1.48, 95% CI: 1.17, 1.88; adjusted OR 2.11, 95% CI: 1.56, 2.86, for males and females, respectively) when compared with their counterparts with higher household income (Table 2). Higher fear of COVID-19 was significantly associated with the RSMC of regular medical visits in 2021 among male and female participants. There was no significant association between low household income and the RSMC of regular medical visits in 2020 (Appendix A).

The odds ratios of equivalent annual income for the RSMC of new symptoms are shown in Table 3. Fear of COVID-19 presented a significant association with the RSMC of new symptoms in 2021, among all the participants. Male participants with relative poverty presented a significantly increased OR for the RSMC of new symptoms in 2021, after adjusting for the covariates. No significant association was observed between relative poverty and the RSMC of new symptoms among female participants in 2021, when compared with their counterparts with higher annual household income. No significant association was observed between income and the RSMC of new symptoms in 2020 among males and females, as provided in Appendix A.

Appendix B summarizes the odds ratios of equivalent annual income in 2020 for the RSMC of regular visits among participants with underlying diseases. Male participants with income below the relative poverty line presented an increased OR for the RSMC of regular visits in crude analysis in 2021, but significance was lost after adjusting for fear of COVID-19 and other covariates. Female participants with income below the relative poverty line were at a significantly increased OR for the RSMC of regular medical visits in 2021, after adjusting for covariates.

Appendix C presents the odds ratios of equivalent annual income for the RSMC of new medical visits among participants with underlying conditions. Significant association was observed between income below relative poverty line and the RSMC of new medical visits among male participants in 2020 (crude analysis only) and 2021, but no significance was observed among female participants.

Analyses according to alert specific areas for COVID-19 are shown in Table 4 (regular medical visits) and Table 5 (new medical visits). Female participants with income below the relative poverty line who were living in the areas subjected to the state of emergency or specific alert prefectures for COVID-19 were more likely to show an RSMC of regular medical visits in 2021 than those with higher income (adjusted OR: 1.79, 95% CI: 1.31, 2.45). Male participants with income below the relative poverty line in alert specific areas for COVID-19 were significantly more associated with the RSMC of new symptoms than their counterparts (adjusted OR: 1.63, 95% CI: 1.22, 2.17). The odds ratios of low household income for the RSMC of regular and new visits in 2020 are provided in Appendix D.

## 4. Discussion

This study showed that household equivalent annual income below the relative poverty line was associated with the RSMC of regular visits and new symptoms during the COVID-19 pandemic, even after adjusting for fear of COVID-19. Our findings also indicated that, while fear of COVID-19 continued to show a significant association with the RSMC of regular visits and new symptoms throughout the study period, low household income became increasingly important for the RSMC of regular visits and new symptoms in 2021. The study added the important finding to the existing literature that financial difficulty due to the COVID-19 pandemic may have led to avoiding behaviors for medical visits.

Previous studies showed lower household income was associated with lower utilization of medical care services prior to the outbreak of the COVID-19 [8,9], as well as during the pandemic [17]. Moreover, studies reported that COVID-19-related concerns have led to avoidance of medical care during the pandemic [20,21] and delayed utilization of medical care [18]. Our findings were in line with the previous studies and added evidence that people with household income below the relative poverty line were at increasing odds for RSMC in 2021, even after adjusting for fear of COVID-19. On the other hand, a previous UK study reported that income equality in primary health care use was present in the first wave of the pandemic but diminished as the COVID-19 pandemic progressed [25], but our finding was contrary to the UK study, indicating the possible need for interventions to eliminate economic disparities in medical care use. Stronger association between low household income and RSMC was observed in the analyses limited to the participants with underlying medical conditions, showing consistence with the previous study that people with underlying diseases were at higher odds for avoiding seeking medical care than those without underlying diseases [26]. As Japan has experienced persistent and consecutive waves of the pandemic in recent years, we assumed that the vulnerable population with lower income may have presented more avoidant behaviors for seeking medical behaviors driven by concerns for medical cost, time, or missing work for making medical visits.

Our analyses showed sex differences in the relationship between low household income and the RSMC of regular and new medical visits. In our study, men with lower income avoided making both regular visits and visits for new symptoms in 2021 more than their counterparts with higher income, while women with lower income showed an RSMC of regular medical visits greater than their counterparts, but not an RSMC of new symptoms. Most previous studies reported higher health service utilization among women than men [10,11,12,13]. On the other hand, a previous study indicated that once women were in the medical system, women received fewer medical services than men with similar health care conditions [27]. These studies seemed to support our findings that men with low household income generally presented an RSMC for both regular visits and new symptoms, and women with low household income attempted to seek medical care for their new symptoms regardless of monetary or time burdens but showed an RSMC for regular medical visits. As our findings showed that low household income and the fear of COVID-19 independently affected avoidant behaviors for making medical visits with no significant interaction term effect between these two variables, different strategies to reduce financial burden (i.e., monetary support for low household income households), as well as to provide medical support without increasing fears for COVID-19 infections (i.e., improving prevention measures against COVID-19 in medical visits, providing online medical support, etc.), may be needed to enhance necessary regular and new medical visits of the vulnerable population in the pandemic.

Our study reported that low household income in 2020 was associated with different types of RSMC in 2021. Missing regular medical visits are associated with the worsening of current medical conditions [28], affecting the severity of underlying diseases and their mortality [29]. People with chronic conditions are at risk for complications when their medical care is not appropriately managed. Our study suggested that low household income affected RSMC among people with underlying conditions during the pandemic, and care coordination for those with underlying diseases was essential. The RSMC of new symptoms, on the other hand, may lead to a delay in early identification and treatment of new and possibly emergent medical conditions. A previous study in Canada reported that there was a significant decline in emergency department visits after the declaration of the COVID-19 pandemic [30]. Our study added to existing evidence that vulnerable populations with low household income were more likely to show an RSMC for new symptoms, which may be presentations of potentially time-sensitive emergencies.

Our study also found significant association between low household income and RSMC among participants living in specific alert areas for COVID-19. It is understandable that people tended to avoid making medical visits when their living areas were under the state of emergency or specific alert for COVID-19. Our findings provided a message that government and healthcare sectors should call out for caution to make necessary medical visits regardless of COVID-19 conditions by providing the evidence that decreased access to medical care puts people at risk for worsening current medical conditions, as well as delaying the treatment for possible diseases.

The study was subject to several limitations. First, the study relied on self-report for household income and refraining from medical visits during the pandemic. Future studies should confirm the current finding with subjective measure for household income and medical access using the data linkage. Second, as our study was conducted using internet surveys, the findings may not be generalizable to the whole population. Our findings should be replicated with a more diverse population in future studies. Third, we did not ask if participants received telemedicine treatment in replacement of their missed medical visits. Fourth, the study might have overestimated the participants not having RSMC, as people who did not seek medical care at the time of the survey might have answered as not having RSMC, possibly leading to the underestimation of association between low household income and RSMC. Finally, our study did not evaluate the health effect resulting from the RSMC of regular or new visits. The impact of RSMC may differ depending on underlying medical conditions and prescriptions. Future studies should explore the long-term effect on health outcomes resulting from RSMC.

## 5. Conclusions

In conclusion, this study showed that household income below the relative poverty line was significantly associated with the RSMC of regular or new visits in 2021, presenting some difference according to sex, underlying diseases, or living areas. The future studies should explore long-term consequences of RSMC, as well as focus on strategies to alleviate economic inequalities in access to medical care in the prolonged pandemic.

## Figures and Tables

**Table 1 ijerph-20-02682-t001:** Characteristics of participants in the study.

	Male (*N* = 10,097)						Female (*N* = 9575)					
	Equivalent Annual Income											
	Below Poverty Line (*N* = 1979)		Above Poverty Line (*N* = 6564)		Unknown (*N* = 1554)		Below Poverty Line (*N* = 2086)		Above Poverty Line (*N* = 5195)		Unknown (*N* = 2294)	
	No	(%)	No	(%)	No	(%)	No	(%)	No	(%)	No	(%)
Age (years)												
20–34	338	(17.0)	1138	(17.3)	269	(17.3)	381	(18.3)	913	(17.6)	345	(15.0)
35–49	466	(23.6)	2013	(30.7)	400	(25.7)	529	(25.4)	1487	(28.6)	621	(27.1)
50–64	459	(23.2)	1852	(28.2)	428	(27.5)	511	(24.5)	1403	(27.0)	701	(30.6)
65–79	716	(36.2)	1561	(23.8)	457	(29.4)	665	(31.9)	1392	(26.8)	627	(27.3)
Marital status												
Married	1157	(58.5)	4531	(69.0)	885	(57.0)	1035	(49.6)	3532	(68.0)	1395	(60.8)
Never married	681	(34.4)	1635	(24.9)	583	(37.5)	585	(28.0)	1057	(20.4)	594	(25.9)
Separated	37	(1.9)	95	(1.5)	27	(1.7)	175	(8.4)	250	(4.8)	127	(23.0)
Divorced	104	(5.3)	303	(4.6)	59	(3.8)	291	(14.0)	356	(6.9)	178	(7.8)
Educational attainment												
Junior high school or high school	647	(32.7)	1438	(21.9)	433	(27.9)	875	(42.0)	1457	(28.1)	793	(34.6)
Technical/junior college/college dropout	274	(13.9)	751	(11.4)	225	(14.5)	676	(32.4)	1682	(32.4)	764	(33.3)
College/university/graduate school	1051	(53.1)	4363	(66.5)	888	(57.1)	529	(25.4)	2048	(39.4)	727	(31.7)
Others/missing	7	(19.6)	12	(65.0)	8	(0.5)	6	(0.3)	8	(0.2)	10	(0.4)
Employment status												
Full or part time	838	(45.3)	5151	(78.5)	848	(54.6)	406	(19.5)	2024	(39.0)	598	(26.1)
Unemployed	199	(10.1)	197	(3.0)	110	(21.7)	500	(24.0)	916	(17.6)	420	(18.3)
Others/missing	942	(47.6)	1216	(18.5)	596	(38.4)	1180	(56.6)	2255	(43.4)	1276	(27.1)
Living areas												
Alert specific areas in 2020 ^a^	432	(21.8)	1356	(20.7)	289	(18.6)	422	(20.2)	1087	(20.9)	481	(21.0)
Other areas in 2020 ^b^	1547	(78.2)	5208	(79.3)	1265	(81.4)	1664	(79.8)	4108	(79.1)	1813	(79.0)
Alert specific areas in 2021 ^c^	964	(48.7)	3485	(53.1)	864	(55.6)	1034	(49.6)	3151	(60.7)	1284	(56.0)
Other areas in 2021 ^d^	1015	(51.3)	2719	(41.4)	690	(44.4)	1052	(50.4)	2044	(39.3)	1010	(44.0)
Underlying diseases												
Hypertension	561	(28.4)	1636	(24.9)	393	(25.3)	390	(18.7)	706	(13.6)	309	(13.5)
Diabetes	252	(12.7)	643	(9.8)	154	(9.9)	95	(4.6)	171	(3.3)	90	(3.9)
Respiratory diseases	66	(3.3)	113	(1.7)	27	(1.7)	51	(2.4)	65	(1.3)	28	(1.2)
Cardiovascular diseases	61	(3.1)	107	(1.6)	26	(1.7)	29	(1.4)	34	(0.7)	12	(0.5)
Stroke	58	(2.9)	97	(1.5)	29	(1.9)	33	(1.6)	46	(0.9)	15	(0.7)
Cancer	69	(3.5)	155	(2.4)	40	(2.6)	53	(2.5)	98	(1.9)	37	(1.6)
Mental disorders	134	(19.6)	180	(2.7)	65	(4.2)	134	(6.4)	143	(2.8)	104	(4.5)
Any	765	(38.7)	2133	(32.5)	509	(32.8)	569	(27.3)	964	(18.6)	476	(20.8)
Fear of COVID-19												
7–15	583	(29.5)	2444	(37.2)	481	(31.0)	504	(24.2)	1462	(28.1)	501	(21.8)
16–20	502	(25.4)	1839	(28.0)	369	(23.8)	651	(31.2)	1741	(33.5)	666	(29.0)
21–25	643	(32.5)	1740	(26.5)	535	(34.4)	616	(29.5)	1417	(27.3)	792	(34.5)
26–35	251	(12.7)	541	(8.2)	169	(10.9)	315	(15.1)	575	(11.1)	335	(14.6)
Refraining from seeking medical care												
Refrained regular medical visit (2020)	193	(9.8)	622	(9.5)	103	(6.6)	276	(13.2)	776	(14.9)	304	(13.3)
(2021)	127	(6.4)	291	(4.4)	46	(3.0)	124	(5.9)	222	(4.3)	119	(5.2)
Refrained medical visit for new symptoms (2020)	79	(4.0)	233	(3.6)	33	(2.1)	112	(5.4)	301	(5.8)	109	(4.8)
(2021)	145	(7.3)	393	(6.0)	76	(4.9)	190	(9.1)	488	(9.4)	182	(7.9)

^a^ Alert specific areas in 2020: restricted areas subject to the state of emergency or specific alert prefectures in April, 2020 (Tokyo, Kanagawa, Chiba, Osaka, Hyogo, Fukuoka, Hokkaido, Ibaragi, Ishikawa, Gifu, Aichi, and Kyoto). ^b^ Other areas were area not subject to emergency alert to COVID-19 in 2020. ^c^ Alert specific areas in 2021: restricted areas subject to the state of emergency in February, 2021 (Tokyo, Kanagawa, Chiba, Saitama, Tochigi, Aichi, Gifu, Osaka, Kyoto, Hyogo, and Fukuoka). ^d^ Other areas were not subject to emergency alert to COVID-19 in 2021.

**Table 2 ijerph-20-02682-t002:** Odds ratios (ORs) of household income for refraining from seeking medical care (RSMC) of regular visits.

Male (*N* = 10,097)						
			Year 2021			
	*N*	Number of Cases (%)	RSMC of Regular Visits			
			Crude OR	95% CI	Adjusted OR ^a^	95% CI
Equivalent annual income						
On or above poverty line	6564	291 (4.4)	1.00 (REF)		1.00 (REF)	
Below poverty line	1979	127 (6.4)	1.28	(1.19–1.83)	1.29	(1.02–1.63)
Unknown or no response	1554	46 (3.0)	0.66	(0.48–0.90)	0.62	(0.45- 0.86)
Fear of COVID-19						
7–15 (low)	3508	133 (3.8)	1.00 (REF)		1.00 (REF)	
26–35 (high)	961	60 (6.2)	1.69	(1.24–2.31)	1.59	(1.15–2.18)
**Female (*N* = 9575)**						
			**Year 2021**			
	** *N* **	**Number of Cases (%)**	**RSMC of Regular Visits**			
			**Crude OR**	**95% CI**	**Adjusted OR ^a^**	**95% CI**
Equivalent annual income						
On or above poverty line	5195	222 (4.3)	1.00 (REF)		1.00 (REF)	
Below poverty line	3086	124 (5.9)	1.42	(1.13–1.78)	1.48	(1.17–1.88)
Unknown or no response	2294	119 (5.2)	1.23	(0.98–1.54)	1.23	(0.97–1.56)
Fear of COVID-19						
7–15 (low)	2467	90 (3.7)	1.00 (REF)		1.00 (REF)	
26–35 (high)	1225	101 (8.2)	2.34	(1.77–3.18)	2.11	(1.56–2.86)

Abbreviations: CI, confidence interval; OR, odds ratio; REF, reference. ^a^ Adjusted for age, educational attainment, employment status, fear of COVID-19, and RSMC in 2020.

**Table 3 ijerph-20-02682-t003:** Odds ratios (ORs) of household income for refraining from seeking medical care (RSMC) of new symptoms.

Male (*N* = 10,097)						
			Year 2021			
	*N*	Number of Cases (%)	RSMC of New Symptoms			
			Crude OR	95% CI	Adjusted OR ^a^	95% CI
Equivalent annual income						
On or above poverty line	6564	488 (9.4)	1.00 (REF)		1.00 (REF)	
Below poverty line	1979	190 (9.1)	1.24	(1.02–1.51)	1.29	(1.02–1.63)
Unknown or no response	1554	182 (7.9)	0.81	(0.63–1.04)	0.62	(0.45–0.86)
Fear of COVID-19						
7–15 (low)	3508	157 (4.5)	1.00 (REF)		1.00 (REF)	
26–35 (high)	961	82 (8.5)	1.99	(1.51–2.63)	1.59	(1.15–2.18)
**Female (*N* = 9575)**						
			**Year 2021**			
	** *N* **	**Number of Cases (%)**	**RSMC of New Symptoms**			
			**Crude OR**	**95% CI**	**Adjusted OR ^a^**	**95% CI**
Equivalent annual income						
On or above poverty line	5195	488 (9.4)	1.00 (REF)		1.00 (REF)	
Below poverty line	2086	190 (9.1)	0.97	(0.81–1.15)	0.94	(0.79–1.14)
Unknown or no response	2294	182 (7.9)	0.83	(0.70–0.99)	0.82	(0.68–0.98)
Fear of COVID-19						
7–15 (low)	2467	152 (6.2)	1.00 (REF)		1.00 (REF)	
26–35 (high)	1225	166 (13.6)	2.39	(1.89–3.01)	2.22	(1.75–2.83)

Abbreviations: CI, confidence interval; OR, odds ratio; REF, reference. ^a^ Adjusted for age, educational attainment, employment status, fear of COVID-19, and RSMC in 2020.

**Table 4 ijerph-20-02682-t004:** Odds ratios (ORs) of household income for refraining from seeking medical care (RSMC) of regular visits according to specific alert areas.

Male (*N* = 10,097)						
			Year 2021			
	*N*	Number of Cases (%)	RSMC of Regular Visits			
			Crude OR	95% CI	Adjusted OR ^a^	95% CI
Alert specific area ^b^:						
Equivalent annual income						
On or above poverty line	3845	163 (4.2)	1.00 (REF)		1.00 (REF)	
Below poverty line	964	65 (6.7)	1.63	(1.21–2.20)	1.35	(0.97–1.87)
Unknown or no response	864	24 (2.8)	0.65	(0.42–1.00)	0.58	(0.37–0.91)
Other areas ^c^:						
Equivalent annual income						
On or above poverty line	2719	128 (4.7)	1.00 (REF)		1.00 (REF)	
Below poverty line	1015	62 (6.1)	1.32	(0.963–1.80)	1.22	(0.87–1.71)
Unknown or no response	690	22 (3.2)	0.67	(0.42–1.06)	0.66	(0.41–1.05)
**Female (*N* = 9575)**						
			**Year 2021**			
	** *N* **	**Number of Cases (%)**	**RSMC of Regular Visits**			
			**Crude OR**	**95% CI**	**Adjusted OR ^a^**	**95% CI**
Alert specific area ^b^:						
Equivalent annual income						
On or above poverty line	3151	120 (3.8)	1.00 (REF)		1.00 (REF)	
Below poverty line	1034	70 (6.8)	1.83	(1.35–2.49)	1.78	(1.30–2.45)
Unknown or no response	1284	71 (5.5)	1.48	(1.09–2.00)	1.48	(1.09–2.02)
Other areas ^c^:						
Equivalent annual income						
On or above poverty line	2044	102 (5.0)	1.00 (REF)		1.00 (REF)	
Below poverty line	1052	54 (5.1)	1.03	(0.73–1.45)	1.17	(0.82–1.66)
Unknown or no response	1010	48 (4.8)	0.95	(0.70–1.35)	0.95	(0.66–1.37)

Abbreviations: CI, confidence interval; OR, odds ratio; REF, reference. ^a^ Adjusted for age, educational attainment, employment status, fear of COVID-19, and RSMC in 2020. ^b^ Alert specific areas in 2021: restricted areas subject to the state of emergency in February, 2021 (Tokyo, Kanagawa, Chiba, Saitama, Tochigi, Aichi, Gifu, Osaka, Kyoto, Hyogo, and Fukuoka). ^c^ Other areas were area not subject to emergency alert to COVID-19.

**Table 5 ijerph-20-02682-t005:** Odds ratios (ORs) of household income for refraining from seeking medical care (RSMC) of new symptoms according to specific alert areas.

Male (*N* = 10,097)						
			Year 2021			
	*N*	Number of Cases (%)	RSMC of New Symptoms			
			Crude OR	95% CI	Adjusted OR ^a^	95% CI
Alert specific area ^b^:						
Equivalent annual income						
On or above poverty line	3845	224 (5.8)	1.00 (REF)		1.00 (REF)	
Below poverty line	964	80 (8.3)	1.46	(1.12–1.91)	1.58	(1.18–2.12)
Unknown or no response	864	43 (5.0)	0.85	(0.61–1.18)	0.99	(0.70–1.40)
Other areas ^c^:						
Equivalent annual income						
On or above poverty line	2719	169 (6.2)	1.00 (REF)		1.00 (REF)	
Below poverty line	1015	65 (6.4)	1.03	(0.77–1.39)	1.06	(0.77–1.46)
Unknown or no response	690	33 (4.8)	0.76	(0.52–1.11)	0.77	(0.52–1.15)
**Female (*N* = 9575)**						
			**Year 2021**			
	** *N* **	**Number of Cases (%)**	**RSMC of New Symptoms**			
			**Crude OR**	**95% CI**	**Adjusted OR ^a^**	**95% CI**
Alert specific area ^b^:						
Equivalent annual income						
On or above poverty line	3151	289 (9.2)	1.00 (REF)		1.00 (REF)	
Below poverty line	1034	115 (11.1)	1.24	(0.99–1.56)	1.18	(0.93–1.49)
Unknown or no response	1284	97 (7.6)	0.81	(0.64–1.03)	0.79	(0.61–1.01)
Other areas ^c^:						
Equivalent annual income						
On or above poverty line	2044	199 (9.7)	1.00 (REF)		1.00 (REF)	
Below poverty line	1052	75 (7.1)	0.71	(0.54–0.94)	0.72	(0.54–0.96)
Unknown or no response	1010	85 (8.4)	0.85	(0.65–1.11)	0.85	(0.64–1.12)

Abbreviations: CI, confidence interval; OR, odds ratio; REF, reference. ^a^ Adjusted for age, educational attainment, employment status, fear of COVID-19, and RSMC in 2020. ^b^ Alert specific areas in 2021: restricted areas subject to the state of emergency in February, 2021 (Tokyo, Kanagawa, Chiba, Saitama, Tochigi, Aichi, Gifu, Osaka, Kyoto, Hyogo, and Fukuoka). ^c^ Other areas were area not subject to emergency alert to COVID-19.

## Data Availability

The data that support the findings of this study are available from the corresponding author, E.O., upon reasonable request.

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
