# Peer review of "Association between Poverty and Refraining from Seeking Medical Care during the COVID-19 Pandemic in Japan: A Prospective Cohort Study"

_ijerph, 2023, doi:10.3390/ijerph20032682_

Round 1
Reviewer 1 Report
The topic is very interesting and the authors have done a reasonable job of conducting the empirical work. However, it has potential to become more interesting.
1. The authors need to present in a clearer way the contribution of this research, given the large body of existing literature on this topic. My question: what is novel? What is the contribution of this paper?
2. The introduction should end with a presentation of the other sections that make up the work.
3. Research questions are not clearly stated.
4. The section “Conclusions”is so simple.
Author Response
Reviewer 1
- The authors need to present in a clearer way the contribution of this research, given the large body of existing literature on this topic. My question: what is novel? What is the contribution of this paper?
We appreciate your important comment. We tried to explain the contribution of the research in the following section (Line 60-65, Page 2):
“Not only will these findings clarify the impact of low household income on RSMC, but al-so give us a clue on who are more vulnerable ones among those with low household in-come, and what type of medical visits will be influenced most. These findings may lead us to draw more specific and practical implications (i.e. providing more monetary or non-monetary support for making medical visit for people with underlying diseases and low income, etc) for overcoming income disparities in health care utilization during the COVID-19 pandemic.”
- The introduction should end with a presentation of the other sections that make up the work.
Thank you very much for your important suggestion. We added a presentation of the other sections in the end of introduction section according to your advice (Line 80-82, Page 2):
“The study then discusses differences in sex, underlying diseases and living areas with or without specific alert on association between low household income on RSMC of regular and new visits.”
- Research questions are not clearly stated.
Thank you for your comment. We added the following sentences to state our research questions following your advice (Line 73-76, Page 2);
“Our research questions are as follows: how does low household income influence RSMC of regular and new visits during the pandemic, even after adjusting for fears of COVID-19? How is the impact of low household income on RSMC different depending on demographic or social factors?”
- The section “Conclusions”is so simple.
Thank you for your advice. We revised the conclusion to provide more detailed description of our study (Line 320-324, Page 14):
“In conclusion, the study showed that household income below relative poverty line was significantly associated with RSMC of regular or new visits in 2021, present-ing some difference according to sex, underlying diseases, or living areas. The future studies should explore long-term consequences of RSMC, as well as focus on strategies to alleviate economic inequalities in access to medical care in the prolonged pandemic.”
Reviewer 2 Report
This study aims to investigate the association between household income below relative poverty line and RSMC of regular visits and new symptoms and the association between the fear of COVID-19 and RSMC of regular and new visits using longitudinal surveys. There a few concerns that I have:
1. I feel that the authors could refrain from using 'low income', which suggests low personal income and instead use 'low household income' because a person could have a high income but needs to support many people in the family and is thus categorised as having low household income. It reduces ambiguity as well.
2. Was the association between fear of COVID-19 and medical care in 2020 or 2021 because line 71 said 2020?
3. There are 3 time-points in line 81 even though the authors mentioned two surveys. Do they mean the first wave of survey was conducted between Aug and Sep 2020 and the second wave in Feb 2021?
4. What does "an invalid response to choose the second alternative from the bottom" (line 93) mean?
5. It's "50% and above" isn't it, not "above 50%" (line 106) since it includes 3 million yen? And it should be "On or above relative poverty line" instead of just "above relative poverty line" else where are the people who are on the poverty line?
6. The way the questions were phrased assumed that people who did not have any regular medical visits within the last 3 months or people who did not have any new symptoms that required medical visits and would then answer "no" to the questions will not refrain from seeking medical care if they need it but it's not 100% true. These people could still refrain from seeking medical care when they need to attend a regular medical visit or had new symptoms that required a medical visit. Shouldn't the authors be comparing among people who need medical care in the last 3 months, how many refrained from seeking medical care vs. how many did not refrain from seeking medical care? Why didn't the authors exclude people who did not need any medical care in the last 3 months?
Furthermore, given that it was the pandemic period, non-urgent medical visits in many countries were postponed due to manpower and resource constraints so it's possible that patients have a 6 monthly medical visit or even a yearly medical visit instead of a 3 monthly medical visit isn't it?
It's also peculiar that the recall period was only 3 months when the survey was conducted almost 6 months apart.
7. How were the covariates selected?
8. (lines 141-143) Are the authors trying to say they did not include interaction terms? If so, please state that interaction terms were not included instead of using lesser known terminology like "product terms" and "main terms".
9. Why wasn't fear of COVID-19 (line 144) adjusted for comorbidities?
10. Instead of using the word "risk" throughout the article, shouldn't the authors be using the word "odds" because they've used a logistic regression to estimate odds ratio so the numbers show an increased or decreased odds, not increased or decreased risk. If the authors wanted to use "risk", they should have estimated the relative risk instead.
Author Response
Reviewer 2
- I feel that the authors could refrain from using 'low income', which suggests low personal income and instead use 'low household income' because a person could have a high income but needs to support many people in the family and is thus categorised as having low household income. It reduces ambiguity as well.
Thank you very much for suggestion. We changed the term ‘low income’ into ‘low household income’ throughout the manscript.
- Was the association between fear of COVID-19 and medical care in 2020 or 2021 because line 71 said 2020?
We appreciate your question. We evaluated the association between fear of COVID-19 and RSMC in both 2020 and 2021, and thus we corrected as the following (Line79-80, Page 2):
“ The study also evaluates the association between the fear of COVID-19 and RSMC of regular and new visits in 2020 and 2021.”
- There are 3 time-points in line 81 even though the authors mentioned two surveys. Do they mean the first wave of survey was conducted between Aug and Sep 2020 and the second wave in Feb 2021?
We corrected the description as follows (Line 91-92, Page 2):
“The two surveys used in the study were conducted in between August and September 2020, and February 2021, respectively.”
- What does "an invalid response to choose the second alternative from the bottom" (line 93) mean?
We are sorry for the confusion. In this question, we asked the respondents to choose the second alternative from the bottom to identify respondents who gave inconsistent responses (those who randomly selected the alternatives). We corrrected the description as following (Line 104-106, Page 3):
“The following criteria were used to assess the inconsistency:an invalid response for not choosing the right alternative as specified (respondents were asked to choose the second option from the bottom);
- It's "50% and above" isn't it, not "above 50%" (line 106) since it includes 3 million yen? And it should be "On or above relative poverty line" instead of just "above relative poverty line" else where are the people who are on the poverty line?
Thank you for your suggestion. We corrected “household income above relative poverty line” as “household income on or above 50% of median income”(Line 118, Page 3).
- The way the questions were phrased assumed that people who did not have any regular medical visits within the last 3 months or people who did not have any new symptoms that required medical visits and would then answer "no" to the questions will not refrain from seeking medical care if they need it but it's not 100% true. These people could still refrain from seeking medical care when they need to attend a regular medical visit or had new symptoms that required a medical visit. Shouldn't the authors be comparing among people who need medical care in the last 3 months, how many refrained from seeking medical care vs. how many did not refrain from seeking medical care? Why didn't the authors exclude people who did not need any medical care in the last 3 months?
Furthermore, given that it was the pandemic period, non-urgent medical visits in many countries were postponed due to manpower and resource constraints so it's possible that patients have a 6 monthly medical visit or even a yearly medical visit instead of a 3 monthly medical visit isn't it?
It's also peculiar that the recall period was only 3 months when the survey was conducted almost 6 months apart.
Thank you very much for your important point. We agree that people who did not refrain from seeking medical visits might have had more less need or symptoms than those who refrained from seeking medical care. We instead provided a supplementary analysis on participants with underlying diseases (Appendix B and C) and found the similar findings with the main analysis. We added the following description in the limitation section (Line 312-314, Page 14):
”Fourth, it was possible that people with more health concerns or symptoms might have refrained from seeking medical visits than those with less health problems.”
We noticed that this survey asked RSMC in the past 2 months, instead of 3 months. We apologize for this confusion. We agree with your point that during the pandemic period, people may have delayed non-urgent medical visits due to the social situation for several months, however, we considered that RSMC in the 2 months period may be an issue especially for people who need more frequent referral or emerging symptoms.
- How were the covariates selected?
We selected the covariates based on the previous stuides and added fears of COVID-19 which was considered to be important during the pandemic. We added the explanation as follows (Line 131-133, Page 3):
“The study used the covariates including variables that were used in the previous stud-ies on RSMC, as well as additional variable (fears of COVID-19) that was considered important during the pandemic.”
- (lines 141-143) Are the authors trying to say they did not include interaction terms? If so, please state that interaction terms were not included instead of using lesser known terminology like "product terms" and "main terms".
Thank you for your suggestion. We corrected as “interaction terms”.
- Why wasn't fear of COVID-19 (line 144) adjusted for comorbidities?
We appreciate your opinion. We did not adjust for comorbidities but instead added an supplementary analysis on participants with comorbidities (Appendix B and C), showing slightly lower odds ratios for RSMC of regular medical visists, and slightly higher odds ratios for RSMC of new symptoms.
- Instead of using the word "risk" throughout the article, shouldn't the authors be using the word "odds" because they've used a logistic regression to estimate odds ratio so the numbers show an increased or decreased odds, not increased or decreased risk. If the authors wanted to use "risk", they should have estimated the relative risk instead.
Thank you for your suggestion. We corrected the term “risk” into “odds” according to your suggestion.
Round 2
Reviewer 2 Report
Regarding point 6, my concern wasn't "people who did not refrain from seeking medical visits might have had more need or symptoms than those who refrained from seeking medical care". My concern was that the authors might have overestimated the number of people who did not refrain from seeking medical care by assuming that those who didn't need any medical care would not refrain from seeking medical care because the survey did not differentiate among these people properly:
a) people who did not need medical care and refrain from seeking care
b) people who need medical care and refrain from seeking care
c) people who did not need medical care and did not refrain from seeking care
d) people who need medical care and did not refrain from seeking care
In other words, people belonging to (a) and (c) were grouped with people belonging to (d) due to the nature of the question.
Regarding point 7, please give the citation of the previous studies.
Author Response
Reviewer 2
Regarding point 6, my concern wasn't "people who did not refrain from seeking medical visits might have had more need or symptoms than those who refrained from seeking medical care". My concern was that the authors might have overestimated the number of people who did not refrain from seeking medical care by assuming that those who didn't need any medical care would not refrain from seeking medical care because the survey did not differentiate among these people properly:
- a) people who did not need medical care and refrain from seeking care
- b) people who need medical care and refrain from seeking care
- c) people who did not need medical care and did not refrain from seeking care
- d) people who need medical care and did not refrain from seeking care
In other words, people belonging to (a) and (c) were grouped with people belonging to (d) due to the nature of the question.
Thank you very much for explaining your point in detail. We are sorry for our misinterpretation of your comment, and we agree that the study might have overestimated the people not having RSMC, as people who did not need medical care at time of the survey would answer as “not having RSMC”, possibly leading to the underestimation of association between low household income and RSMC. We added the following description in the limitation section (Line 308-311, Page 13):
“Fourth, the study might have overestimated the participants not having RSMC, as people who did not medical care at time of survey might have answered as not having RSMC, possibly leading to the underestimation of association between low household income and RSMC.
Regarding point 7, please give the citation of the previous studies.
We added the citation of the previous studies as follows (Line 131-133, Page 3):
“The study used the covariates including variables that were used in the previous studies on RSMC [8, 9, 17, 18, 20, 21], as well as additional variable (fears of COVID-19) that was considered important during the pandemic.”